# Ameliorations in dyslipidemia and atherosclerotic plaque by the inhibition of HMG-CoA reductase and antioxidant potential of phytoconstituents of an aqueous seed extract of *Acacia senegal* (L.) Willd in rabbits

Jaykaran Charan[1], Priyanka Riyad[2], Heera Ram[2]*, Ashok Purohit[2], Sneha Ambwani[1], Priya Kashyap[3], Garima Singh[4], Abeer Hashem[5], Elsayed Fathi Abd_Allah[6], Vijai Kumar Gupta[7], Ashok Kumar[8], Anil Panwar[8]

1 Department of Pharmacology, All India Institute of Medical Sciences, Jodhpur, Rajasthan, India, 2 Department of Zoology, Jai Narain Vyas University, Jodhpur, Rajasthan, India, 3 University School of Biotechnology, GGS Indraprastha University, New Delhi, India, 4 Department of Botany, Pachhunga University College, Aizawl, Mizoram, India, 5 Botany and Microbiology Department, College of Science, King Saud University, Riyadh, Saudi Arabia, 6 Plant Production Department, College of Food and Agricultural Sciences, King Saud University, Riyadh, Saudi Arabia, 7 Center for Safe and Improved Food & Biorefining and Advanced Biomaterials Research Center, SRUC, Kings Buildings, Scotland, United Kingdom, 8 Centre for Systems biology and bioinformatics, Panjab University Chandigarh, Punjab, India

* hr.zo@jnvu.edu.in, baradhr@gmail.com

## Abstract

The assigned work was aimed to examine the capability of phytoconstituents of an aqueous seed extract of *Acacia senegal* (L.) Willd to inhibit HMG-CoA reductase and regression of the atherosclerotic plaque. The chemical fingerprinting of the test extract was assessed by LC-MS/MS. Consequently, the analyses of *in-vitro*, *in-vivo*, and *in-silico* were executed by using the standard protocols. The *in-vitro* assessment of the test extract revealed 74.1% inhibition of HMG-CoA reductase. *In-vivo* assessments of the test extract indicated that treated hypercholesterolemic rabbits exhibited a significant ($P \leq 0.001$) amelioration in the biomarker indices of the dyslipidaemia i.e., atherogenic index, Castelli risk index(I&II), atherogenic coefficient along with lipid profile. Subsequently, significant reductions were observed in the atherosclerotic plaque and antioxidant levels. The *in-silico* study of molecular docking shown interactions capabilities of the leading phytoconstituents of the test extract i.e., eicosanoic acid, linoleic acid, and flavan-3-ol with target protein of HMG-CoA reductase. The values of RSMF and potential energy of top docked complexes were show significant interactions. Accordingly, the free energy of solvation, interaction angle, radius of gyration and SASA were shown significant stabilities of top docked complex. The cumulative data of results indicate phytoconstituents of an aqueous seed extract of *Acacia senegal* have capabilities to inhibit the HMG-CoA reductase and improve the levels of antioxidants.

**Data Availability Statement:** All relevant data are within the paper and its Supporting information files.

**Funding:** Researchers Supporting Project Number (RSP-2021/134), King Saud University, Riyadh, Saudi Arabia.

## Introduction

The existing therapeutics of dyslipidemia involve cholesterol lowering drugs specifically known as statins and fibrates. The mechanism of statins involves inhibition of HMG-CoA enzyme [1]. Although, there are several adverse effects associated with these synthetic drugs [2]. In view of this, the present study was aimed to explores HMG–CoA reductase inhibition and antioxidant potential. Plant products are not only used in traditional medicine but are also in demand globally as potential sources for the development of new drugs [3]. The Indigenous traditional herbal remedies contain unique formulations of local herbs and herbal extracts that have been developed based on conventional knowledge and local wisdom [4–6]. The ability of several traditional medicines to treat and resolve cardiovascular problems and linked meta-bolic disorders have been well documented [7]. In this regard, polyherbal formulation of five local herbs (Panchkuta), such as unripe pod of *Prosopis cineraria* (Sangari), seed of *Acacia sen-egal (L.)* Willd. (Kumbat or Kumatiya), fruit of *Capparis decidua* (Ker), fruit of *Cordia myxa* (Gunda), and pulp of unripe fruit of *Mangifera indica* (Amchoor) that are endemic to the Western Rajasthan region (Thar desert) of India, have been historically used to treat cardiovas-cular problems in rural communities [8, 9]. The seeds of *Acacia senegal* (L.) Willd. is one of the key ingredients in this herbal medicine (panchkuta) of which several medicinal properties have been demonstrated in our previous studies [10–12]. Exudates of *Acacia senegal*(L.) Willd., which is commonly known as gum Arabic, have also been reported to exhibit hypocho-lesterolemic activity in animals as well as Sudanese human subjects [13–15]. The extracts of the seeds of *Acacia senegal* (L) also have the ability to inhibit serine proteinase activity [16]. Several reports have provided the information about the ethnopharmacological applications of foods and herbal medicines of indigenous to the arid regions of African countries and the Indian subcontinent [16–18]. *Acacia senegal* (L.) Willd.is typically known by its common name, white gum tree, and is a member of the Leguminosae-Mimosoideae [11, 19], while seed extracts of *Acacia senegal* (L.) Willd.is locally known as kumbat or kumatiya in Rajasthan [20, 21]. The present study also identified the major phytoconstituents present in the seed extracts of *Acacia senegal* and assess its anti-atherosclerotic properties in hypercholesterolemic rabbits using a combination of *in-vitro*, *in-silico*, and *in-vivo* methodology.

## Material and methods

### Plant material and extraction

The seeds of *Acacia senegal* (L.) Willd. were collected from in and around premises of new campus of Jai Narain Vyas University, Jodhpur (Rajasthan), India. Taxonomic confirmation of the seeds was based on a comparison with an herbarium accession by a botanical expert in the regional centre, Botanical Survey of India, Jodhpur (BSI/AZRC/I.12012/Tech./2021-22 (PI. Id.)/007 dated 16.06.2021). Seed extract was obtained using a standard Soxhlet procedure [22].

### Identification of the phytoconstituents

The screening of predominant phytoconstituents present in the seed extracts was based on LC-MS (Liquid chromatography and Mass spectroscopy) [23, 24]. The LC-MS data were sub-sequently analysed using Mass hunter software developed by Agilent. Peaks generated in both positive and negative modes of ionization, with $\geq$3500 ionization counts, were considered using a peak spacing tolerance of 0.0090m/z for reasonable resolution of the chromatogram. Chromatogram peaks were assigned masses based upon MS-MS fragmentation patterns spe-cific for the identified phytocompound. The metabolite profile was confirmed using mass Bank workstation software along with public database information. The samples (SAIF 436)

were analysed by the SAIF (Sophisticated Analytic Instrumental Facility), CDRI, Lucknow, UP, India.

## Chemicals and reagents

All chemicals and reagents were used obtained from Sigma Aldrich, India up to chemical grade of ACS (American Chemical Society). Diagnostic kits were obtained from local supplier of Transasia Bio-Medicals LTD, Erba Mannheim GmbH., Germany.

## Doses of standard statin drug and seed extract dosage

A supply of 20 mg tablets of Atorlip (atorvastatin) was obtained from a local pharmacy in Jodhpur and administered doses were calculated based on body weight of the test rabbits. The 400mg/kg dose regime was calculated and administered orally for the course of experimentation based on $LD_{50}$ assessment and previously published studies [25, 26].

## *In-vitro* inhibition of HMG -CoA reductase activity

The HMG-CoA reductase inhibition assay was performed *in-vitro* using a kit (Sigma Aldrich) according to the manufacturer's instructions and previous reports in the literature [27, 28]. The inhibitory activity of increasing concentrations (0.32μg/ml, 0.62 μg/ml, 1.25 μg/ml, and 5μ0g/ml) of the seed and a standard statin drug (Pravastatin) provided with the kit were determined by measuring absorbance at 340 nm. The $IC_{50}$ was calculated based on the obtained inhibition curve for HMGR of the seed extract and the standard drug. The assay is based on the decrease in absorbance resulting from the tested compound and measures the oxidation of NADPH by the catalytic subunit of HMGR in the presence of the substrate HMG-CoA.

## Experimental animals

New Zealand white male adult rabbits weighing approximately 1.5±0.1 kg were used in the experiments. Four groups (two control groups and two treated groups) of rabbits were formulated by consisting of five rabbits in each group. Animals were acclimatized for 10 days prior to the onset of the experiment and were maintained in cages in a controlled environment (26 ± 3˚C and 12 h of light and dark cycles). The animals were fed a balanced diet supplemented with micronutrients and vitamins. The experimental protocol for use of the animals was recommended (UDZ/IAEC/2019/03 dated on 29.03.2019) by the Institutional Animal Ethics Committee (IAEC) based on the standard norms of the CPCSEA (Reg. No.1646/GO/a/ 12/CPCSEA valid up to 27.03.23).

Experimental groups were assigned as follows:

Group I: Intact control

Group II: Hypercholesterolemic control

Group III: Group administered seed extracts of *Acacia senegal* (L.) Willd.

Group IV: Group administered standard statin drug (Atorvastatin).

The duration of the experiment was 60 days inclusive of the time needed to induce hypercholesterolemia (15days) and administer the treatments (45days). After the completion of experimentation, the overnight fasted animals were scarified after cervical dislocation by flowing the guidelines of AVMA (The American Veterinary Medical Association) [29].

### Induction of hypercholesterolemia

Hypercholesterolemia was induced in the test rabbits by feeding them a high fat diet and a cholesterol powder supplement for 15days. The cholesterol powder supplement was formulated at 500mg cholesterol powder/kg body weight per day mixed with 5ml coconut oil [30, 31]. The induction of hypercholesterolemia was confirmed by weekly biochemical assessments of the blood lipid profile and calculation of the atherogenic index using standard methods.

### Collection of serum samples for biochemical and histopathological analyses

Twenty-four-hour fasted animals were autopsied under prolonged anaesthesia of ketamine formulation (10mg/kg) as per recommendation of the veterinarian at the completion of the experiment and blood samples were obtained from direct cardiac and hepatic vein puncture. The collected blood was kept in EDTA-coated vials and serum was separated by centrifugation for 15 min at 3000rpm.

### Serum lipid profile and atherogenic index

Total cholesterol [32], HDL-cholesterol [33], and triglyceride (TG) [34] were determined using standard methods and the lipid profile was constructed following Friedewald's formula [35]. The following indices were calculated using the indicated formulas:

$$LDL\text{-}cholesterol = Total\ cholesterol\ \text{-}\ HDL\text{-}cholesterol\ \text{-}\ VLDL\text{-}cholesterol$$

Where VLDL = triglyceride/5

The Castelli risk index–I (Total cholesterol/HDL), Castelli risk index–II (LDL/HDL) [36] and the Atherogenic index = Log (Triglyceride / HDL-cholesterol) [37].

### Antioxidants and peroxidation assays of serum

Serum antioxidant levels were determined for catalase [38], superoxide dismutase (SOD) [39], GSH (reduced glutathione) [40], and FRAP (Ferric reducing antioxidant potential) [41] using standard protocols based on redox reaction end products measured as absorbance at an appropriate wavelength. The degree of lipid peroxidation (LPO) in serum was determined by assessing thiobarbituric acid reactive substances (TBARS) and is represented as malondialdehyde (MDA) content, following the modified method of Ohkawa [42].

### Histology and planimetric (morphometry) study of aorta

A 2–3 cm length of the ascending aorta of autopsied animals was removed and fixed in 10% formalin. The aortic tissues were consequently dehydrated through alcohol series and eventually implanted in paraffin wax. The paraffin-embedded samples of aorta were sectioned at a thickness of 5 microns and processed for staining and histopathological analysis [10, 43]. The morphometric measurements and planimetric assessments of the sectioned samples of aorta were performed using a Camera Lucida [30, 43].

### *In-silico* assessments

*In-silico* assessments were performed by following the molecular docking, molecular dynamics simulation, ADMET and pharmacokinetics.

## Molecular docking

The interaction compatibility of the screened prominent phytocompounds with HMG-CoA reductase (1HW8) was examined through the molecular docking by using Autodock 4 [44, 45]. The catalytic portion of human HMG-CoA reductase (1HW8) was retrieved from a protein data bank and managed using PyMol to obtain the co-crystallised ligand i.e., atorvastin, eliminate undesirable water molecules, and correct for chain integration. Three-dimensional structures of the compounds identified in the seed extract and the known inhibitors (pravastatin and atorvastatin) were downloaded from Pubchem Database. Ligand processing was performed using PyMol and hydrogen atoms were added to the structures. The developed docking protocol was validated by performing re-docking with prepared co-crystalized ligand and composed receptor protein and maps were created. Post-validation of the docking etiquette of the test compounds was performed by independently docking them with target receptor proteins. The parameters of molecular interactions were obtained through ligand conformations, binding energies, and linked assessments.

## Molecular dynamics

Molecular dynamics (MD) simulation assessments were conducted by using GROMACS to recognize the conformational dynamics of docked complexes (Atorvastatin, Eicosanoid, Flavan-3-ol, Linoleic acid and Pravastatin) with 1HW8. The MD simulations of docked complexes such as atorvastatin-HMG-CoA reductase (1HW8), Eicosonoid-1HW8, Flavan-3-ol-1HW8, Linoleic acid-1HW8 and Pravastatin-1HW8 were performed with the GROMACS 2020 [46]. For the solvation of protein, dodecahedron box was used, and protein was placed at least 1.0 nm from the edge of the box. The standard protocol and conditions were followed by structural analysis (RMSF and potential energy minimization) of top three docked complexes which were further proceeded for top docked complex (Eicosonoid-1HW8) was made by using radius of gyration, free energy of solvation, average angle, angle distribution, SASA and interaction energy by gyrate modules of GROMACS and their representations (curves) were produced with xm grace (Graphing, Advanced Computation and Exploration program).

## ADMET pharmacokinetic analysis

The pharmacokinetics of ADMET analyses were performed using Drulito software with the standard protocol used to determine the ideal pharmacokinetic profile of the test compounds considered for drug development [47–49]. The test compounds were adopted through two filters: the Lipinski rule and the blood brain barrier (BBB) requirement. The Lipinski rule indicates that an ideal drug molecule should weigh below 500g/mol, the number of hydrogen bond donors should be less than or equal to 5 and the number of hydrogen bond acceptor should be ≤ 10, with a partition coefficient ≤ 5. The test compound should pass the BBB if the number of hydrogen bonds present is approximately 8–10 and no acidic groups should be present in the molecule. TPSA (total polar surface area) represents the bioavailability of the drug molecule according to Veber's rule which indicates that a TPSA less than or identical to 140Å will have good oral bioavailability.

## Statistical analysis

The data on the biochemical parameters were represented as a mean ± SEM (standard error of the mean). A one-way analysis of variance (ANOVA) was conducted followed by Tukey's multiple comparison tests using GraphPad Prism 7.0 software. Graphical representations of the data were constructed using MS Excel 2018.

**Table 1. Identified masses from UPLC-QTOF mass spectroscopy constituents in an aqueous extract of *Acacia. senegal* (L.) Willd. seed in negative and positive electron ionization modes.**

| S.No. | Identified compound Name | Formula | Monoisotopic mass (g/mol) | Retention time (min) | m-z/ m+z values |
|---|---|---|---|---|---|
| 1. | Fisetinidol | $C_{15}H_{14}O_5$ | 274.1 | 1.05 min | 273.1 |
| 2. | Linoleic acid | $C_{18}H_{32}O_2$ | 280.4 | 2.31 min | 279.4 |
| 3. | Eicosonoic acid | $C_{20}H_{40}O_2$ | 312.02 | 3.84 min | 311.09 |
| 4. | Lupenone | $C_{30}H_{48}O$ | 424.5 | 23.00 min | 423.5 |
| 5. | Flavan-3-ol | $C_{15}H_{14}O_2$ | 226.04 | 4.47 min | 249.0 |
| 6. | Myricetin | $C_{15}H_{10}O_8$ | 318.3 | 4.58 min | 341.3 |
| 7. | Digallic acid | $C_{14}H_{10}O_9$ | 322.2 | 13.10 min | 323.2 |
| 8. | Taxifolin | $C_{15}H_{12}O_7$ | 304.3 | 16.23 min | 327.3 |
| 9. | Gallocatechin | $C_{15}H_{14}O_7$ | 306.3 | 16.23 min | 307.3 |

## Results

### Phytoconstituents screening by LCMS

The monoisotopic mass obtained for phytoconstituents was calculated as M+H or M-H ions in QTOF mass hunter software and verified by MS/MS and identified using the data MET-LINE software and published literature. Results indicated that the seed extract contained nine major phytoconstituents (Table 1).

### *In-vitro* inhibition of HMG-CoA reductase activity

The seed extract and the standard statin drug, pravastatin, exhibited a maximum 74.1% and 91.4% inhibition of HMG-CoA reductase activity, respectively. Increasing gradient of concentrations of the seed extract were assessed. Enzyme activity was calculated based on the product rate per minute. The $IC_{50}$ of the seed extract, calculated from the inhibition curve, was 0.064μg/ml (Fig 1A & 1B).

### Atherogenic index, Castelli risk indexes (I &II), and the lipid profile

Biomarker indices of dyslipidemia, such as atherogenic index, Catelli risk index–I (Total cholesterol/HDL), Castelli risk index–II (LDL/HDL), and the lipid profile significantly (P≤ 0.001) increased up to ten-fold, relative to the vehicle group, in rabbits that were fed the high fat diet supplemented with cholesterol powder. Treatment with the seed extract or atorvastatin resulted in a significant reduction in the atherogenic index, LDL/HDL ratio, and lipid profile that were near normal relative to the untreated rabbits (Figs 2 & 3).

### Effect on peroxidation and antioxidants levels

The levels of peroxidation and antioxidants (SOD, CAT and GSH) were abnormal in hypercholesterolemic rabbits. In contrast, however, administration of the seed extract or atorvastatin resulted insignificant reduction (P≤ 0.001) in MDA in hypercholesterolemic rabbits, relative to the untreated, hypercholesterolemic rabbits. Moreover, the levels of catalase, SOD and GSH were significantly elevated in hypercholesterolemic rabbits which treated with the test seed extract. Increased levels of total antioxidants were observed in the rabbits treated with the seed extract, as determined by using a FRAP assay (Fig 4).

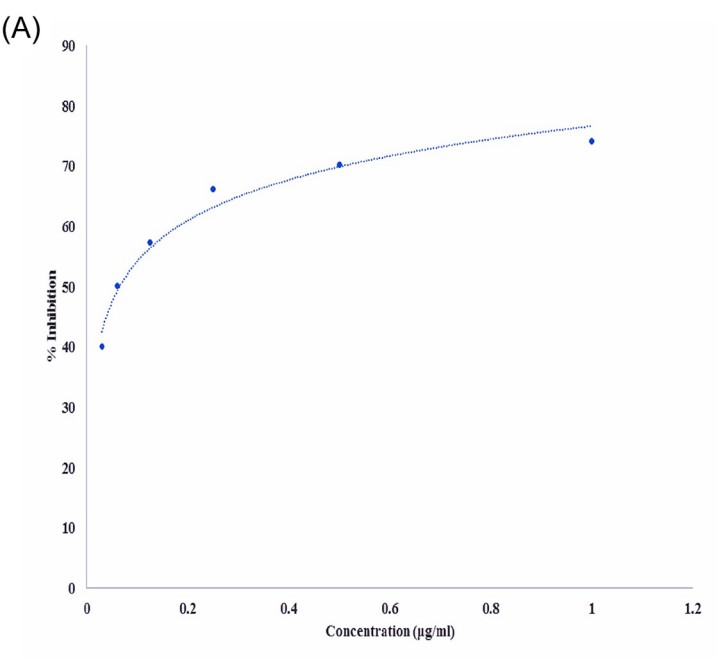

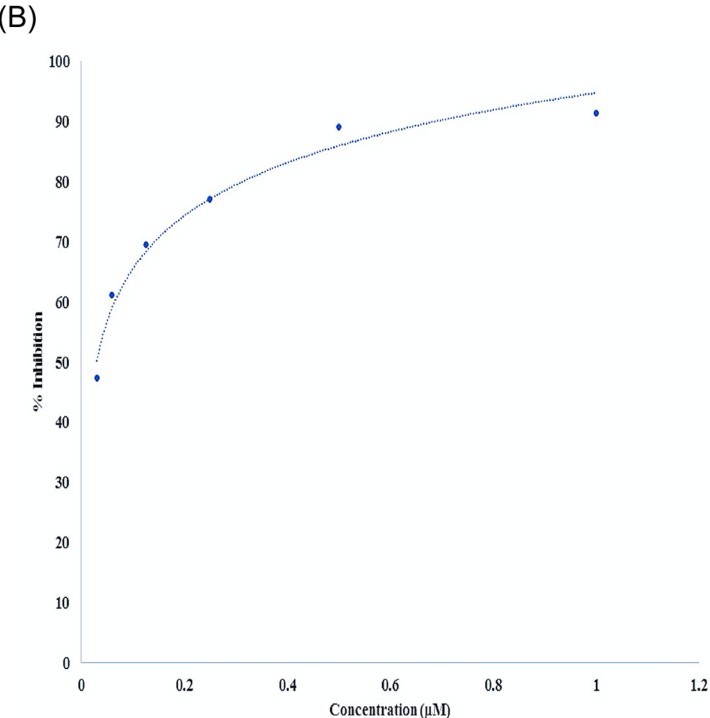

**Fig 1. A:** HMG-CoA reductase inhibition against ascending concentration gradient of the aqueous extract of *Acacia. senegal* (L.) Willd. seed (Equation- y = 9.7365ln(x) + 76.671, $R^2$ = 0.9725, IC50 = 0.064μg/ml). **B:** HMG-CoA reductase inhibition against ascending concentration gradient of the standard drug (Pravastatin) (Equation-y = 12.686ln(x) + 94.755, $R^2$ = 0.9749, IC50 = 0.029μM).

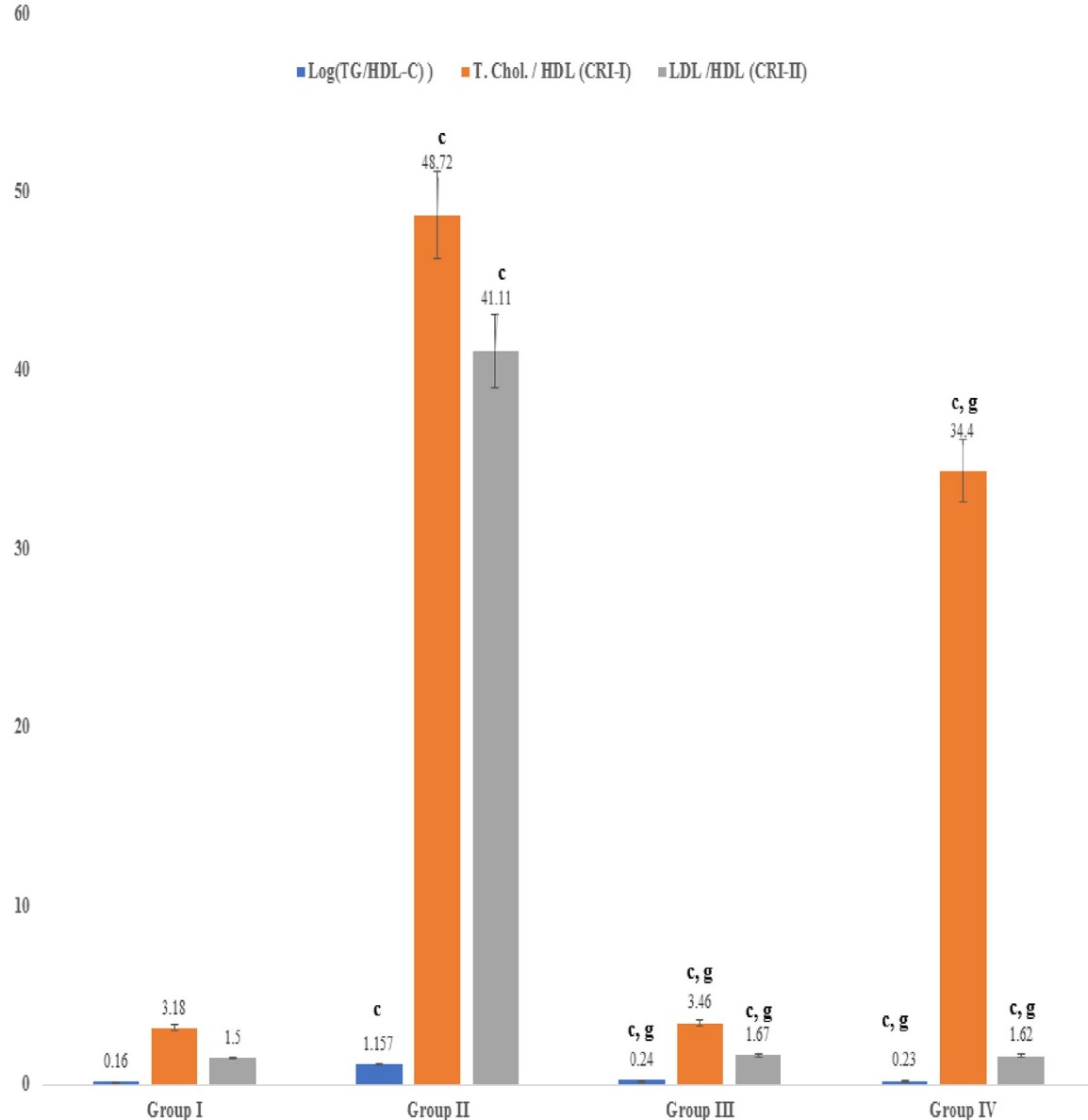

**Fig 2. Effect on biomarker indices of dyslipidemia i.e., Castelli risk factors (I & II) and atherogenic index (AI) of phytochemicals of an aqueous extract of *Acacia. senegal* (L.) Willd. seed.** Data are means ± S.E.M. (n = 5); a $P \leq 0.05$; b $P \leq 0.01$; c $P \leq 0.001$; and d was non-significant as compared to the respective control values. e $P \leq 0.05$; g $P \leq 0.001$; and h was non-significant as compared to the respective values of the hypercholesterolemic control group.

## Histology and morphometric (planimetric) analysis of the aorta

The aortal wall of the vehicle control group (non-hypercholesterolemic) of rabbits was composed of three distinct layers (intima, media and adventitia) and exhibited a compact wall area and enlarged lumen (Figs 5 and 6A). In contrast, the aortal wall of hypercholesterolemic rabbits exhibited abnormal wall area with the presence of bulging structures of atherogenic plaque and a reduced lumen volume (Figs 5 and 6B). Treatment of the hypercholesterolemic rabbits with the seed extract resulted in a significant (P≤ 0.001) reduction in the aortal total wall area and plaque along with an enlargement in lumen volume relative to the untreated,

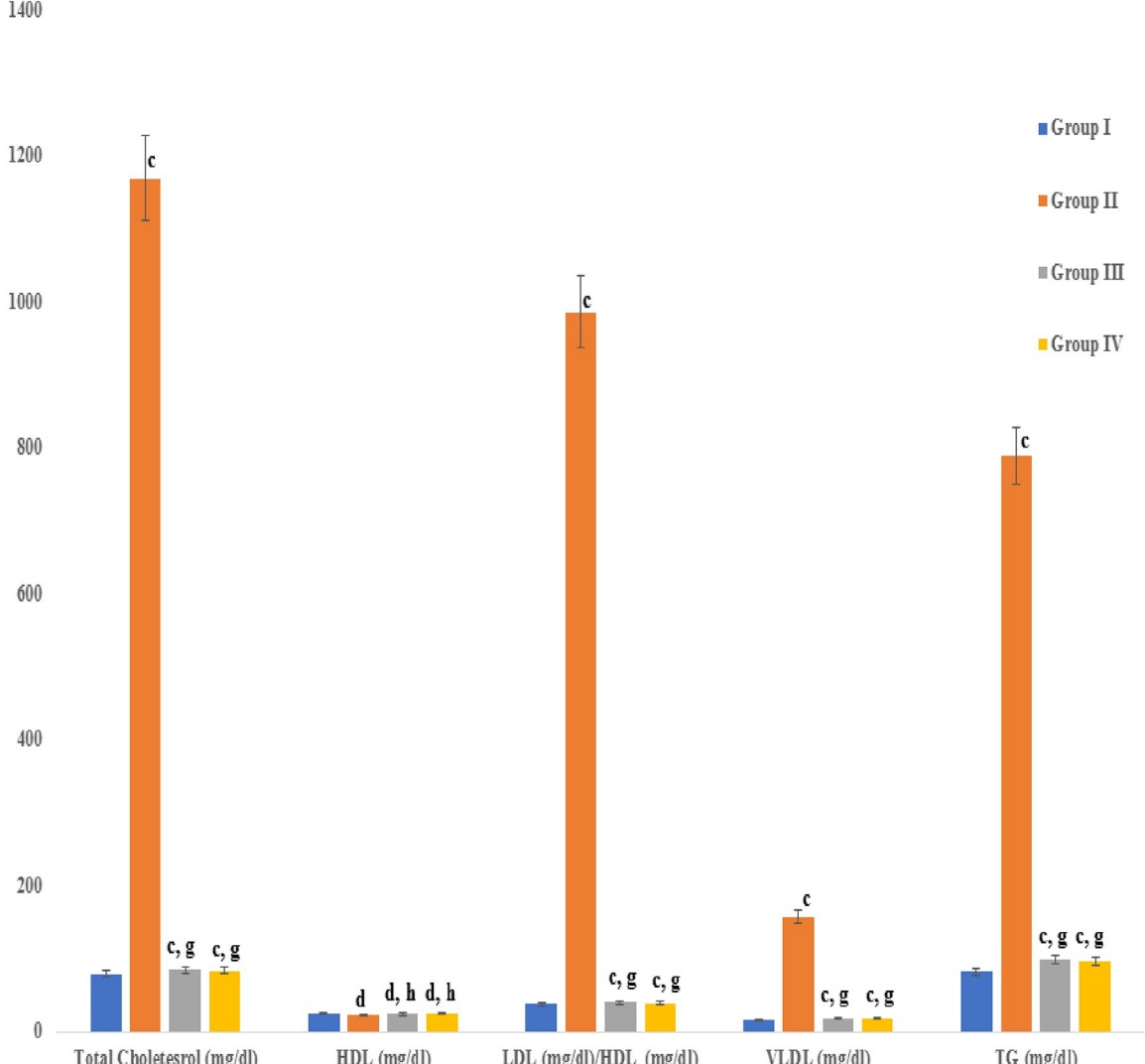

**Fig 3. Effect of an aqueous extract of *Acacia. senegal* (L.) Willd. seed treatment on lipid profile.** Data are means ± S.E.M. (n = 5); a $P \leq 0.05$; b $P \leq 0.01$; c $P \leq 0.001$; and d was non-significant as compared to the respective control values. e $P \leq 0.05$; g $P \leq 0.001$; and h was non-significant as compared to the respective values of the hypercholesterolemic control group.

hypercholesterolemic rabbits. The effect was even greater than the reduction exhibited in response to treatment with the standard statin drug (Figs 5, 6C & 6D).

### *In-silico* assessments

The *in-silico* assessments were performed by following the assessments of molecular docking, molecular dynamics, ADMET and Pharmacokinetics through standard procedures where results obtained as followings.

### Molecular docking

HMG-CoA has a catalytic groove comprising amino acid residue from 426 to 888. The catalytic portion is composed of Cys688, Thr689, Asp690 and Lys691. The side chain of Lys691 is

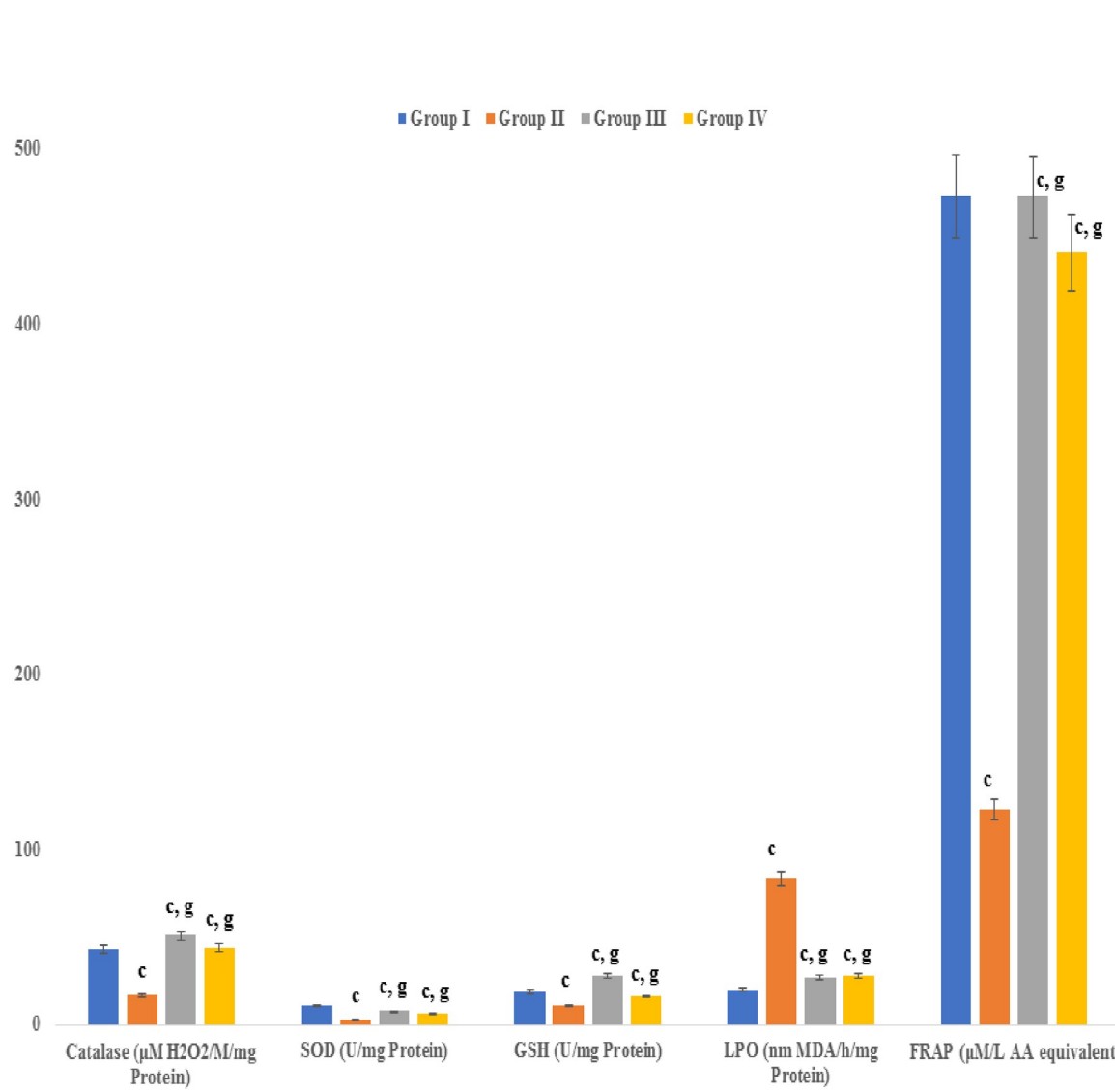

**Fig 4. Effect of an aqueous extract of *Acacia. senegal* (L.) Willd. seed on antioxidant levels in treatment groups.** Data are means ± S. E.M. (n = 5); a $P \le 0.05$; b $P \le 0.01$; c $P \le 0.001$; and d was non-significant as compared to the respective control values. e $P \le 0.05$; g $P \le 0.001$; and h was non-significant as compared to the respective values of the hypercholesterolemic control group.

positioned in the middle of the active site. The flap, primarily composed of Glu559 and Asp767, is in the front of the active site. Among the identified phytoconstituents, eicosanoic acid, linoleic acid, digallic acid, and flavan-3-ol displayed polar interactions with the catalytic residues of the receptor protein (Table 2, Fig 7A–7E). In contrast, gallocatechin, taxifolin, and myricetin did not exhibit any interaction with the HMG-CoA molecule.

## Molecular dynamics

Atorvastatin_1HW8, Eicosanoid_1HW8, Flavan-3-ol_1HW8, Linoleic acid_1HW8 and Pravastatin_1HW8 protein systems were solvated and made electro neutral by adding 3 sodium

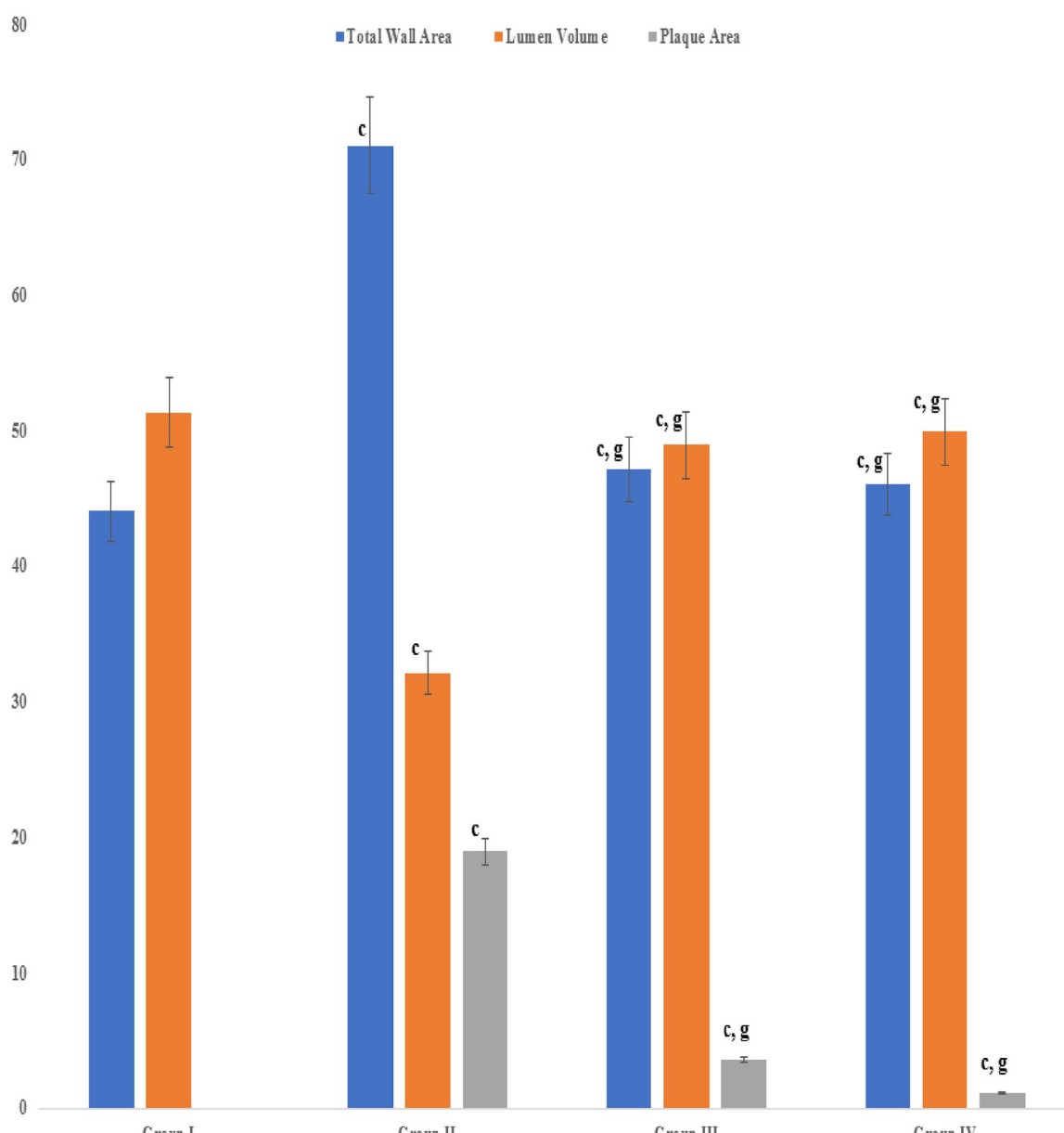

**Fig 5. Effect of an aqueous extract of *Acacia. senegal* (L.) Willd. seed on planimetry of aorta.** Data are means ± S.E.M. (n = 5); a
$P \leq 0.05$; b $P \leq 0.01$; c $P \leq 0.001$; and d was non-significant as compared to the respective control values. e $P \leq 0.05$; g $P \leq 0.001$; and h
was non-significant as compared to the respective values of the hypercholesterolemic control group.

ions in each system using genion module of GROMACS. Potential energy graph revealed a
sudden drop in potential energy of the system in first few ps but reached a constant value
thereafter. Potential energy minimization of the Atorvastatin_1HW8 system achieved at 2060
EM steps, Eicosanoid_1HW8 at 1681 EM steps, Flavan-3-ol_1HW8 at 1873 steps, Linoleic
acid_1HW8 at 1768 steps and Pravastatin_1HW8 at 1754 EM steps, indicating that the struc-
ture Eicosonoid_1HW8 equilibrated fastest among all five (Fig 8A). Appropriately, RMSF per
residue were calculated which shows fluctuation over all the course of study of all residues of
all five proteins. Peak shows protein area undergoing maximum fluctuation over simulation.

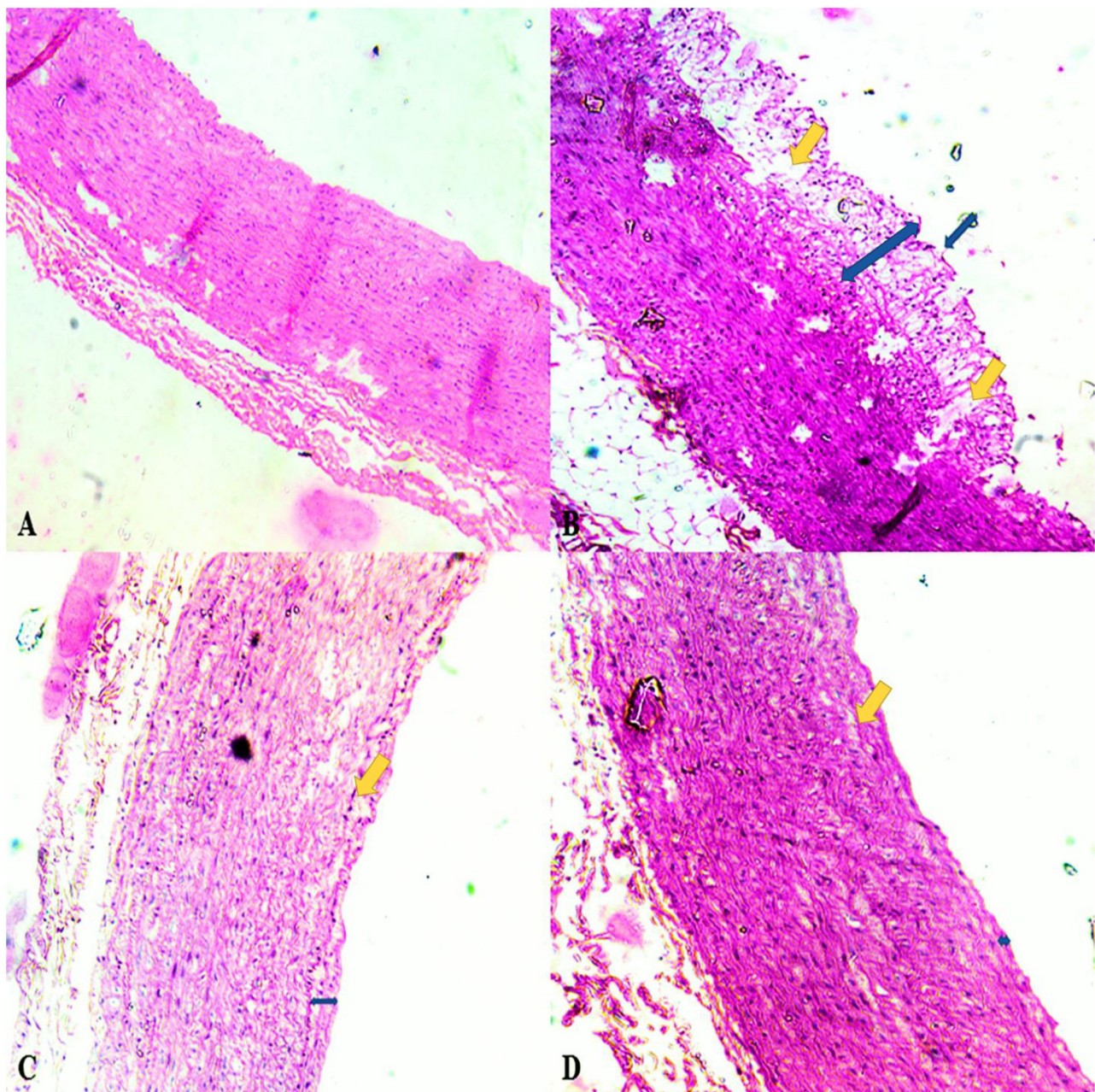

**Fig 6.** Effect of an aqueous extract of *Acacia. senegal* (L.) Willd. seed on histopathology of aortas of treatments groups (400X, H& E), **A–Histoarchitecture of vehicle control aorta:** Exhibiting normal structure with composition three layers i.e. intima, media and adventitia, **B–Histoarchitecture of hypercholesterolemia aorta:** The arrow indicating the presence of atherosclerotic plaque whereas yellow arrow indicating the foam cells in the area of intima, **C- Histoarchitecture of *Acacia. senegal* (L.) Willd. seed extract (aqueous) treated aorta:** The arrow indicating the reduced area of atherogenic aorta, **D-Histoarchitecture of atorvastatin treated aorta:** Histoarchitectural restorations by treatment of atorvastatin by indicating the arrow.

**Table 2. Molecular docking investigations of identified phytocompounds of aqueous extract of _Acacia senegal_ (L.) Willd. seed with target enzyme of HMG-CoA reductase.**

| S.No. | Ligand | Binding Energy (Kcal/mol) | No. of H-bonds | Bond length (Å) | Interacting residues |
|---|---|---|---|---|---|
| Identified Phytoconstituents | | | | | |
| 1. | Fisetinidol | 0.8 | | | |
| 2. | Linoleic acid | -3.4 | 3 | 2.7, 2.5, 2.1 | Asp767, Asp690, Lys692 |
| 3. | Eicosonoic acid | -5.0 | 3 | 3.3, 2.4, 2.4 | Asp690, Lys691, Glu559 |
| 4. | Lupenone | NA | | | |
| 5. | Flavan-3-ol | -3.4 | 1 | 2.4 | Arg590 |
| 6. | Myricetin | 7 | NA | NA | NA |
| 7. | Digallic acid | -3.7 | 9 | 2.7, 2.3, (2.8, 1.8, 2.2), (1.8, 2.6), 2.2, 2.3 | Lys692, Ala751, Lys691, Asn7555, Ser684, Arg590 |
| 8. | Taxifolin | 0.9 | 0 | | |
| 9. | Gallocatechin | 7.5 | NA | NA | NA |
| Positive control | | | | | |
| 1. | Pravastatin | -7.0 | 2 | 1.8, 2.1 | Asp690, Lys691 |
| 2. | Atorvastatin | -7.8 | 1 | 2.2 | Asp690 |

Pravastatin System high RMSF values indicates its residues were more fluctuating during simulation as compared to others Fig 8B. Subsequently, there were seen significant interaction values of ligand interaction energy, average angle, angle distribution and radius of gyration of top docked complex (Eicosanoid_1HW8) (Fig 8C). Supportively, the ligand accessibilities and stabilities shown significant values of free energy of solvation and SASA (Fig 8D).

## ADMET analysis of pharmacokinetics

ADMET studies of the identified phytoconstituents indicated that, among the identified phytoconstituents in the seed extract, only the flavonoid, flavan-3-ol, conforms to the Lipinski rule of five along with the potential to cross the BBB. Although eicosanoic acid and linoleic acid both displayed a molecular interaction with HMG-CoA in the docking analysis, they did not conform with the Lipinski rule of five for an ideal drug molecule. Fisetinidol and taxifolin exhibited ideal drug profiles but lack the ability to cross the BBB and did not interact with the target protein in the docking analysis (Table 3).

## Discussion

The prevailing strategy for the management of hypercholesterolemia is the use of HMG-CoA reductase inhibitors which work by inhibiting cholesterol synthesis by HMG-CoA reductase in the liver and removal of excess cholesterol level in peripheral circulation by several mechanisms of reverse cholesterol transport [50, 51]. Excess cholesterol in the circulatory system is indicated by biomarker indices of dyslipidaemia and abnormal lipoproteins ratios, which can be regulated by proper fractional esterification of cholesterol and reverse cholesterol transport (RCT) [52, 53]. Cholesterol present in the intestine is first absorbed in the form of chylomicron (triglyceride rich complex) and is then modified and packaged as high-density lipoprotein (HDL) cholesterol. Therefore, the ratio of triglyceride to HDL is indicative of the levels of peripheral cholesterol in circulation. Abnormal cholesterol esterification rates in apoB-lipoprotein-depleted plasma (fractional esterification) and lipoprotein particle size result in dyslipidaemia [52, 54]. In animal model, specifically hypercholesteraemic rabbits, exhibit elevated levels of the biomarker indices of dyslipidaemia, such as the logarithm of the TG/HDL ratio, total cholesterol/ HDL (Castelli risk index -I (CRI-I)) and LDL-cholesterol/HDL-cholesterol

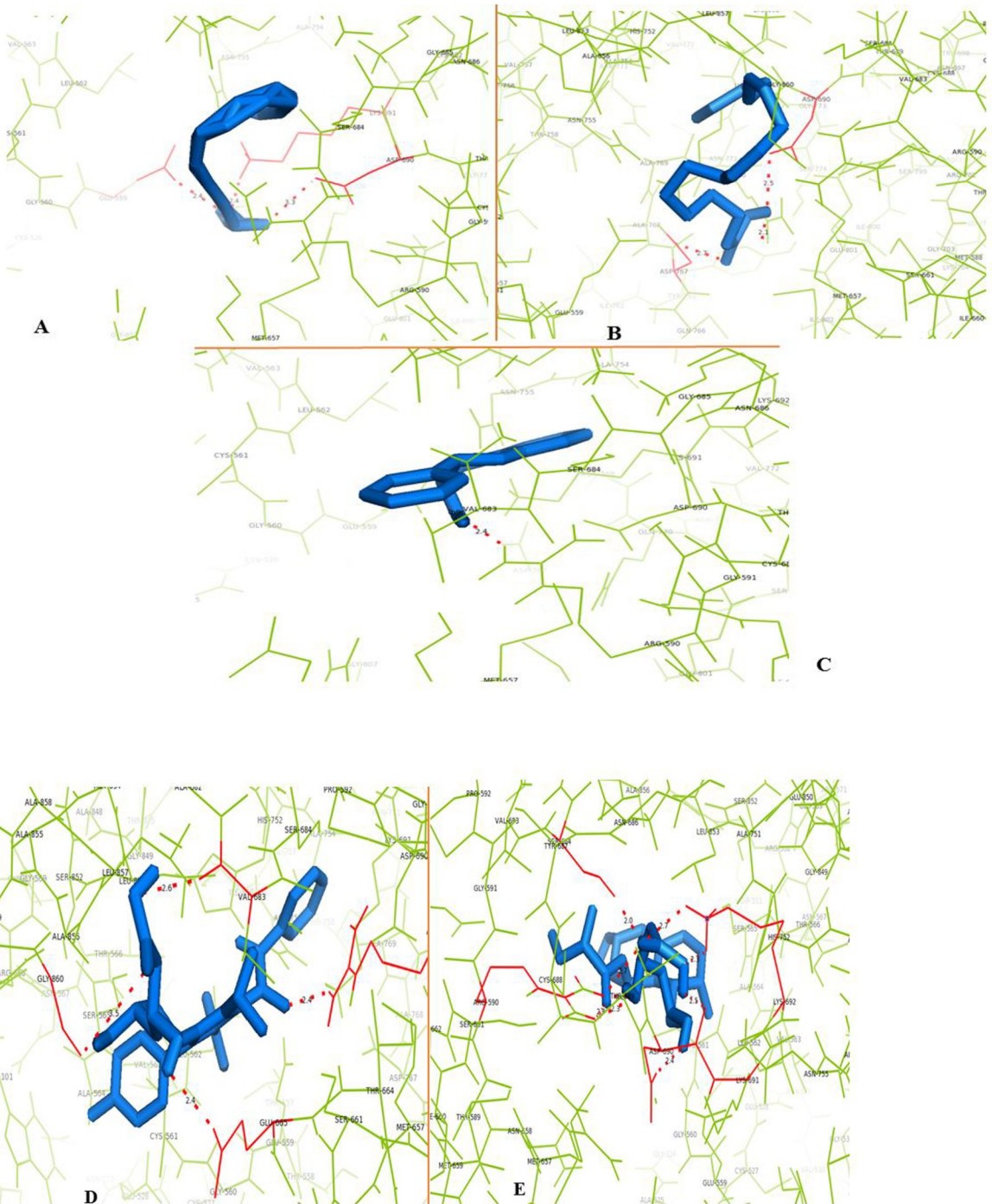

**Fig 7. Molecular interactions of identified compounds studied using docking analysis; (A)- HMG-CoA interaction with eicosonoic acid; (B)- HMG-CoA interaction with linoleic acid; (C)- HMG-CoA interaction with flavan-3-ol; (D)- HMG-CoA interaction with atorvastatin; (E)- HMG-CoA interaction with pravastatin.**

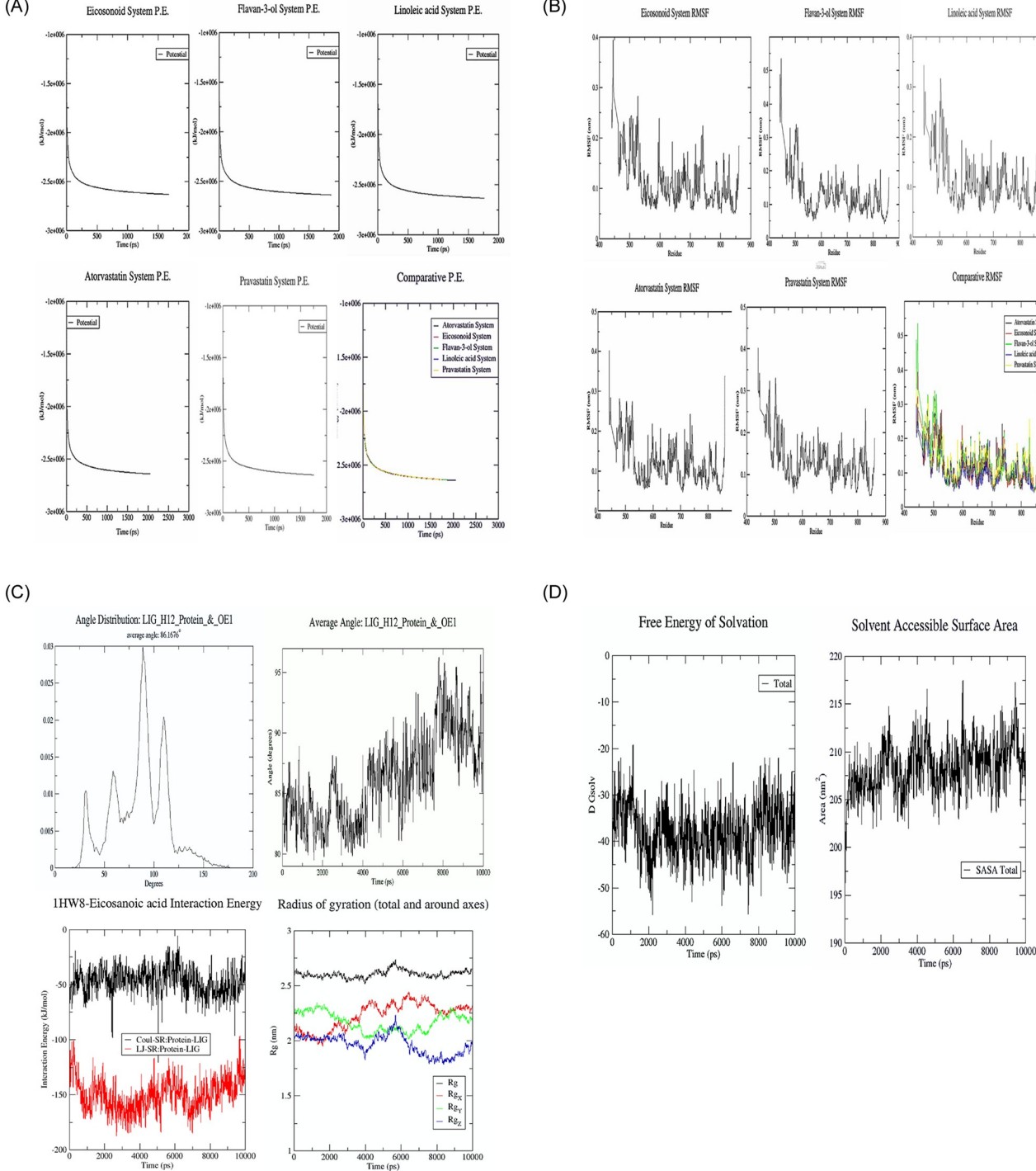

**Fig 8. A:** Potential Energy Minimization of Eicosanoid System achieved at 1681 P.E steps, Flavan-3-ol System achieved at 1873 P.E steps, Flavan-3-ol System achieved at 1873 P.E steps, atorvastatin system achieved at 2060 P.E. steps, Pravastatin System achieved at 1754 P.E steps and Comparative Potential Energy Minimization all five Systems. **B:** System RMSF accounted during the 1ns of MD simulations run of Eicosanoid, flavan-3-ol, linoleic acid, atorvastatin, pravastatin and comparative. **C:** The top docked complex of 1HW8-Eicosanoic acid, showing the highest binding affinity, was subjected to molecular dynamics simulations. The molecular dynamic simulations were examined based on Interaction energy, Free energy of solvation (DGsolv), Radius of gyration (Rg), average angle and angle distribution of ligand in receptor's active site as a function of time. **D:** Molecular dynamics simulation of free energy of solvation and SASA (Solvent Accessible Surface Area) of top docked complex of 1HW8 (target Protein) -Eicosanoic acid.

**Table 3. Pharmacokinetics ADMET prediction by Drulito against Lipinski rule of five and blood-brain-barrier filter of phytocompounds of aqueous extract of *Acacia. senegal* (L.) Willd. seed.**

| Compound | MW | logP | AlogP | HBA | HBD | TPSA | nHB | nAcidic group | Filter L/B |
|---|---|---|---|---|---|---|---|---|---|
| Fisetinidol | 274.08 | 0.933 | -0.373 | 5 | 4 | 90.15 | 9 | 0 | L |
| Linoleic acid | 280.24 | 7.865 | -0.948 | 2 | 1 | 37.3 | 3 | 1 | |
| Eicosonoic acid | 312.3 | 9.846 | -5.05 | 2 | 1 | 37.3 | 3 | 1 | |
| Lupenone | 424.37 | 11.294 | 3.801 | 1 | 0 | 17.07 | 1 | 0 | |
| Flavan-3-ol | 226.1 | 1.591 | 1.316 | 2 | 1 | 29.46 | 3 | 0 | L/B |
| Myricetin | 318.04 | 2.182 | -1.807 | 8 | 6 | 147.68 | 14 | 0 | |
| Digallic acid | 322.02 | 1.77 | -1.178 | 9 | 6 | 164.75 | 15 | 1 | |
| Taxifolin | 304.06 | 0.803 | -1.369 | 7 | 5 | 127.45 | 12 | 0 | L |
| Gallocatechin | 306.07 | 1.2 | -1.499 | 7 | 6 | 130.61 | 13 | 0 | |

MW = molecular weight; logP = partition coefficient; AlogP = octanol–water partition coefficient; HBA = hydrogen bond acceptor; HBD = hydrogen bond donor; TPSA = total polar surface area; nHB = number of hydrogen bond; nAcidic group = number of acidic group; Filter L = Lipinski rule of five and B = blood brain barrier.

(Castelli risk index-II (CRI-II)). In the present study, the treatment of hypercholesterolemic rabbits with an aqueous seed extract of *Acacia senegal* (L.) Willd. caused a significant reduction in the atherogenic index and CRI–I&II, indicating improved fractional esterification of cholesterol and reverse cholesterol transport. These results are similar to a previously reported study [36]. The lipid profile i.e., total cholesterol, triglyceride, VLDL-cholesterol, and LDL-cholesterol were significantly improved by treatment with the aqueous seed extract of *Acacia senegal* (L.) Willd. The seed extract appears to significantly inhibit cholesterol biosynthesis in hepatic tissues, as demonstrated in the *in-vitro* HMG-CoA reductase inhibition assay, as well as the *in vivo* studies in hypercholesterolemic rabbits. A variety of phytocompounds have been reported to have capacity to inhibit HMG-CoA reductase, a key enzyme in cholesterol biosynthesis, by inducing the activation of sterol regulatory element binding protein-2 (SERBP-2) and modifications in LDL receptors that lead to reduced cholesterol production and other parameters of the lipid profile [51, 55].

Excessive amounts of peripheral LDL-cholesterol induce the generation of an excessive level of free radicals resulting in oxidative stress. This causes endothelial dysfunction and leads to the further progression of atherosclerotic plaque and reduced lumen volume in the aorta. Similar observations have been noted hypercholesterolemic animals accompanied by an excess level of cholesterol in the peripheral circulatory system, as well as the progression of atherosclerosis. In the present study, hypercholesterolemic rabbits treated with the seed extract exhibited lower levels of free radicals and elevated levels of catalase, SOD and GSH, which are responsible for scavenging and degrading free radicals. In addition, treatment with the seed extract also resulted in a significant regression in atherosclerotic plaque which would have reversed the progress of atherosclerosis. Previous studies have indicated that hypercholesterolemia promotes atherosclerosis by generating oxidative stress which causes an imbalance between host antioxidant capability and the level of oxidative stress-inducing molecules including reactive oxygen (ROS), nitrogen (RNS), and halogen species, non-radical as well as free radical species. Oxidative stress leads to peroxidation of cellular proteins, lipids, and DNA, resulting in cell injury or cell death, which activates cell death signalling pathways that are responsible for accelerating atherogenesis [56]. In the present study, treatment of hypercholesterolemic rabbits with the seed extract elevated the levels of catalase, SOD and GSH and thus the free radical scavenging capacity of the cell. This effect reduced the atherogenic plaque area and increased the lumen volume. Consequently, Oxidative stress govern through imbalance between free

radicals formation and their antioxidant status (scavenging process) in the body. In the case of hypercholesterolemia, there is an raised level of total cholesterol pool in cells which results into altered cell membrane due to lipid peroxidation [57]. Natural and synthetic antioxidants have been reported to play a crucial role in the prevention and treatment of atherosclerosis through different mechanisms, including inhibition of LDL oxidation [58], decreasing the generation of ROS [59], inhibition of cytokine discharge, the regression of atherosclerotic plaque formation [60] and platelet accumulation [61], the prevention of mononuclear cell infiltration, improvement in endothelial dysfunction [56] and vasodilation, increasing nitric oxide (NO) bioavailability [62], modulating the expression of adhesion molecules, and reducing foam cell formation [61]. The phytochemical analysis of the seed extract identified several predominant phytoconstituents, including fisetinidol, linoleic acid, eicosanoic acid, lupenone, flavan-3-ol, myricetin, digallic acid, taxifolin, and gallocatechin. The *insilico* molecular docking analysis indicated that eicosanoic acid, linoleic acid, and flavan-3-ol are capable of binding to the target enzyme, HMG-CoA reductase [63]. Accordingly, the molecular dynamics (MD) simulation validates the stability of the complex system in polar solution was observed using the parameters of RMSD (root mean square deviation), RMSFs (root means square fluctuations), and radius of gyration [64]. The MD Simulations are very helpful in identifying potential flavonoids and potent ligands targeting disease therapy [65]. A constant trend for RMSF was observed in systems. The protein region between amino acid residues 450–500 shows the highest root mean square fluctuations in all five systems indicating this area of highly dynamic in nature [45].

The average Coulomb's short-range (Coul-SR) value for complex 1HW8-Eicosanoic acid was found -46.75 KJ/mol, indicating that 1HW8-Eicosanoic acid interaction is favourable. The average Lennard-Jones short-range (LJ-SR) value for the complex was found -149.97 KJ/mol. The solvation-free energy of the complex remains static with an average value of -35 DGsolv. Theoretically, the solvent-accessible surface area (SASA) gives an insight into how accessible a protein is to the solvent it resides. Throughout the simulations, SASA fluctuates around 210 $nm^2$ for 1HW8-Eicosanoic acid complex [45, 66]. A plot of the radius of gyration (Rg) spanning over 10 ns is analysed to display the compactness of the protein during MD simulations. Throughout simulations, the radius of gyration for the 1HW8-Eicosanoic acid complex fluctuates around 2.6 nm, indicating that the complex remains stable during simulation studies [67]. MD analysis has revealed that Eicosanoic acid has lesser binding energy, higher nonbonded interaction capability, and more stability against HMG-CoA reductase compared to other ligands. Eicosanoic acid was determined to be the best candidate phytochemicals of an aqueous seed extract of Acacia senegal (L.) against HMG-CoA reductase inhibition. Compassionately, the ADMET profile of the major phytoconstituents present in the seed extract indicated that the compounds have ideal pharmacokinetic properties conforming to the Lipinski rule, have good bioavailability, and are capable of crossing the blood brain barrier [47, 68].

## Conclusion

In conclusion, it can be stated that leading phytoconstituents of an aqueous seed extract of *Acacia senegal* (L.) Willd. i.e., eicosanoic acid, linoleic acid, and flavan-3-ol, have capability to inhibit the HMG-CoA reductase and significantly able to scavenge free radicals. These properties might be responsible to regress atherosclerosis and reduce hypercholesterolemia as evident by the improvements in biomarker indices of dyslipidaemia observed *in vivo* in hypercholesterolemic rabbits. The further efficacy of leading phytocompounds can be validating by alone or in formulation at targeted gene expressions.

## Supporting information

**S1 Fig. A:** QTOF analyses of aqueous extract of Acacia. senegal (L.) Willd. seed extract. **B:** QTOF analyses of aqueous extract of Acacia. senegal (L.) Willd. seed extract. **C:** QTOF analyses of aqueous extract of Acacia. senegal (L.) Willd. seed extract. **D:** QTOF analyses of aqueous extract of Acacia. senegal (L.) Willd. seed extract.
(DOCX)

**S2 Fig. A:** System Temperature graphs of Eicosonoid, flavan-3-ol, linoleic acid, atorvastatin and pravastatin after temperature minimization for 100 picoseconds. **B:** System Pressure graphs of Eicosonoid, flavan-3-ol, atorvastatin, pravastatin and comparative view accounted after NPT Equilibration for 100ps. **C:** System Density graphs of linoleic acid, Eicosonoid, flavan-3-ol, atorvastatin, pravastatin and comparative view accounted after Equilibration for 100ps. **D:** System RMSD of Eicosonoid, Flavan-3-ol, linoleic acid, Pravastatin and atorvastatin. Crystal Backbone (black) Equilibrated Structure Backbone (red). **E:** Radius of gyration for Eicosonoid, Flavan-3-ol, linoleic acid, atorvastatin and pravastatin system accounted after 1000 ps.
(DOCX)

## Acknowledgments

This research did not receive any specific grant from funding agencies in the public, commercial, or not-for-profit sectors. Authors are also acknowledged and considered the contribution of late Prof. Suresh Kumar suffered casualty by the COVID-19 pandemic.

## Author Contributions

**Conceptualization:** Jaykaran Charan, Heera Ram, Sneha Ambwani.

**Data curation:** Ashok Purohit, Garima Singh, Vijai Kumar Gupta.

**Funding acquisition:** Elsayed Fathi Abd_Allah.

**Investigation:** Jaykaran Charan, Priyanka Riyad, Priya Kashyap, Anil Panwar.

**Methodology:** Priyanka Riyad, Anil Panwar.

**Project administration:** Heera Ram.

**Resources:** Ashok Purohit.

**Software:** Priya Kashyap, Ashok Kumar.

**Supervision:** Heera Ram, Sneha Ambwani, Ashok Kumar.

**Validation:** Anil Panwar.

**Writing – original draft:** Heera Ram, Garima Singh.

**Writing – review & editing:** Abeer Hashem, Elsayed Fathi Abd_Allah, Vijai Kumar Gupta, Ashok Kumar.

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
