## [Decision Letter · Decision Letter 0]

24 Jan 2022

PONE-D-21-34360Ameliorations in the biomarker indices of dyslipidemia and atherosclerotic plaque by the inhibition of HMG-CoA reductase and antioxidant potential of phytoconstituents of an aqueous seed extract of Acacia senegal (L.) Willd in rabbitsPLOS ONE

Dear Dr. Ram,

Thank you for submitting your manuscript to PLOS ONE. After careful consideration, we feel that it has merit but does not fully meet PLOS ONE’s publication criteria as it currently stands. Therefore, we invite you to submit a revised version of the manuscript that addresses the points raised during the review process.

We look forward to receiving your revised manuscript.

Kind regards,

Pankaj Kumar Arora

Academic Editor

PLOS ONE

Journal Requirements:

Reviewers' comments:

Reviewer's Responses to Questions

**Comments to the Author**

1. Is the manuscript technically sound, and do the data support the conclusions?

Reviewer #1: Yes

2. Has the statistical analysis been performed appropriately and rigorously? 

Reviewer #1: Yes

3. Have the authors made all data underlying the findings in their manuscript fully available?

Reviewer #1: Yes

4. Is the manuscript presented in an intelligible fashion and written in standard English?

Reviewer #1: Yes

5. Review Comments to the Author

Reviewer #1: The manuscript is well organized and quiet interesting can be accepted after resolving the following minor queries.

1. Title need to short and modify.

2. Page11, Line106 need to modify.

3. Page 11 Line 114, typographic error, “--The inhibitory activity of increasing concentrations (0.32�g/ml, 0.62 �g/ml, 1.25 114 �g/ml, and 5�0g/ml)…”

4. Methodology section should be updated by latest references.

5. Dyslipidemia indices should be explained properly and its roles in therapeutics of hypercholesterolemia.

6. What are the independent and combined roles of dyslipidemia indices such as the Castelli indices (I & II), atherogenic coefficient and atherogenic risk indices?

7. The molecular dynamics of interactions should be validated by SASA or structural analyses.

8. Discussion should be explained by the correlation between the antioxidant properties and hypocholesterolemic activities of the phytocompounds.

9. Histopathology of aorta should be marked with presence of foam cells and regressed area.

10. Discussion should be revised with the explanations of molecular dynamics and molecular docking.

11. Give the explanation of lipid peroxidation and hypercholesterolemia.

12. The druggability evidence should be updated and explained properly with latest references.

6. PLOS authors have the option to publish the peer review history of their article (what does this mean?). If published, this will include your full peer review and any attached files.

Reviewer #1: No

---

## [Author Response · Author response to Decision Letter 0]

3 Feb 2022

Editor-In-Chief, Date: 28.01.2022

PLOS One

Subject: Submission of point-to-point responses against the comments. 

Dear Sir/Madam, 

 I am submitting herewith point to point responses against the comments of submitted revised manuscript (PONE-D-21-34360) entitled “Ameliorations in the biomarker indices of dyslipidemia and atherosclerotic plaque by the inhibition of HMG-CoA reductase and antioxidant potential of phytoconstituents of an aqueous seed extract of Acacia senegal (L.) Willd in rabbits.” in prescribed format by incorporating your suggestions in your esteemed journal. I hope that the paper conforms to the rigorous standards of your journal, and I look forward to hearing your comments and decision concerning the publication of our paper. Please acknowledge and consider for publication. 

Thanking you, 

Yours Sincerely,

(Dr. Heera Ram)

 Corresponding Author 

SN Comments Response 

1. Title need to short and modify. Revised.

2. Page11, Line106 need to modify. Modify.

3. Page 11 Line 114, typographic error, “--The inhibitory activity of increasing concentrations (0.32�g/ml, 0.62 �g/ml, 1.25 114 �g/ml, and 5�0g/ml)…” Revised.

4. Methodology section should be updated by latest references. Updated by latest references as per suggestions.

5. Dyslipidemia indices should be explained properly and its roles in therapeutics of hypercholesterolemia. Explanation incorporated.

6. What are the independent and combined roles of dyslipidemia indices such as the Castelli indices (I & II), atherogenic coefficient and atherogenic risk indices? There are indices of packing of free cholesterol and reverse cholesterol transport pathway.

7. The molecular dynamics of interactions should be validated by SASA or structural analyses. Incorporated.

8. Discussion should be explained by the correlation between the antioxidant properties and hypocholesterolemic activities of the phytocompounds. Explanations incorporated.

9. Histopathology of aorta should be marked with presence of foam cells and regressed area. Suggestion incorporated.

10. Discussion should be revised with the explanations of molecular dynamics and molecular docking. Discussion updated as per suggestions.

11. Give the explanation of lipid peroxidation and hypercholesterolemia. In hypercholesterolemia, there is an increase of total cholesterol pool in cells which results into altered cell membrane due to lipid peroxidation

12. The druggability evidence should be updated and explained properly with latest references. Updated and explanation incorporated.

(Dr. Heera Ram)

 Corresponding Author

---

## [Decision Letter · Decision Letter 1]

15 Feb 2022

Ameliorations in the biomarker indices of dyslipidemia and atherosclerotic plaque by the inhibition of HMG-CoA reductase and antioxidant potential of phytoconstituents of an aqueous seed extract of Acacia senegal (L.) Willd in rabbits

PONE-D-21-34360R1

Dear Dr. Ram,

We’re pleased to inform you that your manuscript has been judged scientifically suitable for publication and will be formally accepted for publication once it meets all outstanding technical requirements.

Kind regards,

Pankaj Kumar Arora

Academic Editor

PLOS ONE

Additional Editor Comments (optional):

Reviewers' comments:

Reviewer's Responses to Questions

**Comments to the Author**

1. If the authors have adequately addressed your comments raised in a previous round of review and you feel that this manuscript is now acceptable for publication, you may indicate that here to bypass the “Comments to the Author” section, enter your conflict of interest statement in the “Confidential to Editor” section, and submit your "Accept" recommendation.

Reviewer #1: All comments have been addressed

2. Is the manuscript technically sound, and do the data support the conclusions?

Reviewer #1: Yes

3. Has the statistical analysis been performed appropriately and rigorously? 

Reviewer #1: I Don't Know

4. Have the authors made all data underlying the findings in their manuscript fully available?

Reviewer #1: Yes

5. Is the manuscript presented in an intelligible fashion and written in standard English?

Reviewer #1: Yes

6. Review Comments to the Author

Reviewer #1: The authors have been responded to comments and suggestions were also incorporated in the revised manuscript with justification.

7. PLOS authors have the option to publish the peer review history of their article (what does this mean?). If published, this will include your full peer review and any attached files.

Reviewer #1: No

---

## [Editor Report · Acceptance letter]

22 Feb 2022

PONE-D-21-34360R1 

Ameliorations in dyslipidemia and atherosclerotic plaque by the inhibition of HMG-CoA reductase and antioxidant potential of phytoconstituents of an aqueous seed extract of *Acacia senegal* (L.) Willd in rabbits 

Dear Dr. Ram:

I'm pleased to inform you that your manuscript has been deemed suitable for publication in PLOS ONE. Congratulations! Your manuscript is now with our production department. 

Kind regards, 

on behalf of

Dr. Pankaj Kumar Arora 

Academic Editor

PLOS ONE